# Comparative efficacy of different types of acupuncture as adjuvant therapy on carotid atherosclerosis: a protocol for systematic review and network meta-analysis

Xianming Wu [1], Qian Mo,[2] Zhihong Yang,[2] Xiaolou Huang,[2] Jiao Liu,[2] Shuangmei Xu,[2] Ning Zhang,[2] Xiaofang Yang[1,2]

XW and QM contributed equally.

¹School of Acupuncture-Moxibustion and Tuina, Hunan University of Chinese Medicine, Changsha, Hunan, China
²School of Acupuncture-Moxibustion and Tuina, Guizhou University of Traditional Chinese Medicine, Guiyang City, Guizhou, China

**Correspondence to**
Ning Zhang;
zhangning_nico@163.com and
Xiaofang Yang;
Yangxiaofang210@163.com

## ABSTRACT

**Introduction** Carotid atherosclerosis (CAS) is a disease of the aorta caused by lipid metabolism disorders and local inflammation. Acupuncture combined with traditional western medicine (such as aspirin or atorvastatin) for the treatment of CAS has been widely applied in clinical practice, but there is still a lack of supporting evidence for its efficacy and safety on CAS. Therefore, this systematic review and network meta-analysis (NMA) will summarise the effects of different types of acupuncture treatments on CAS, and a ranking of the therapeutic classes will also be presented, aiming to provide evidence-based medicine for its extensive clinical application.

**Methods and analysis** Systematic and NMA searches will be conducted in seven electronic databases: PubMed, EMBASE, Medline, Cochrane Library, Chinese National Knowledge Infrastructure, Wanfang Database and Chongqing VIP databases. The search time is from their inception to December 2020, regardless of language and publication type. Randomised controlled trials and controlled clinical trials that include patients with CAS receiving acupuncture therapy compared with a control group will be considered eligible. The primary outcomes include the carotid intima-media thickness and vessel plaque quantification; the secondary outcomes include the carotid plaque Crouse score, greyscale median, lipid levels, the incidence of cardiovascular events, safety and adverse events. The selection of studies, data extraction, quality assessment and risk of bias assessment will be conducted by two independent reviewers. The NMA will be analysed with Stata V.15.0, RevMan V.5.3 software and WinBUGS V.1.4.3.

**Ethics and dissemination** Ethical approval will not be required for this study as it will be based on de-identified, aggregated published data. We will publish the findings in a peer-reviewed journal.

**PROSPERO registration number** CRD42020207260.

### Strengths and limitations of this study

► This study will be the first comprehensive analysis of the efficacy of acupuncture as an adjuvant therapy in the treatment of atherosclerosis. A ranking of the therapeutic classes will also be presented.
► The data report will follow the Preferred Reporting Items for Systematic Reviews and Meta-analyses guidelines.
► This study only includes English and Chinese trials, which may lead to the potential risk of ignoring some studies.
► The heterogeneity of different studies may affect the final results of this study.

thrombosis, intimal injury, inflammatory response, oxidative stress and activation of vascular smooth muscle cells. It is manifested by cell metabolism disorder and product accumulation, which causes the vessel wall to harden and thicken, and lose its elasticity; the inner diameter of the lumen becomes smaller, blood flow is blocked, and finally, the diseased blood vessel blood supply organs and even life are endangered.[1–3] The WHO reports that non-communicable diseases cause 41 million deaths each year, equivalent to 71% of the total global deaths. Among them, cardiovascular diseases cause 17.9 million deaths each year.[4] Globally, the number of people dying from cardiovascular disease is still increasing. The WHO plans to reduce premature mortality from non-communicable diseases by one-third by 2030 through treatment and prevention.[4] Among them, carotid AS (CAS) has common risk factors and pathological basis with AS of the coronary arteries of the heart, cerebral blood vessels and renal arteries.[5–7] A study has found that 93% of cerebral

## INTRODUCTION

Atherosclerosis (AS) is a series of pathological changes in the vascular intima through lipid infiltration, platelet activation,

infarctions are accompanied by CAS, and 18%–25% of ischaemic strokes can be attributed to thromboembolism caused by CAS.[8] Carotid intima-media thickness (cIMT) is related to the risk of cardiovascular events in the general population, and cIMT measurement is essential for assessing the risk and incidence of cardiovascular disease.[9] Of course, vessel plaque quantification (VPQ) is a non-invasive evaluation method for observing the morphological characteristics of plaques, and it is also regarded as the main observation in clinical practice.[10]

The current conventional western medicine treatments for carotid AS mainly consist of lifestyle modification, drug therapy, etc. Lifestyle modification includes smoking cessation, control of energy intake, etc. The drug therapy includes hypolipidaemic and antiplatelet drugs (such as aspirin or atorvastatin).[11] Statins can treat AS by regulating blood lipids, improving vascular endothelial function, inhibiting thrombosis, anti-inflammatory and anti-oxidation, and stabilising plaque.[12] Acupuncture is part of the characteristic treatment of traditional Chinese medicine. Its main feature is to stimulate the meridians and acupoints and

regulate the qi and blood of the viscera to achieve the purpose of disease prevention and treatment. It has the effects of anti-inflammation, immunity activation and nervous system modulation.[13 14] At present, acupuncture is widely used as an adjuvant treatment of AS as a characteristic treatment method of Chinese medicine.[15 16] One of the mechanisms of electroacupuncture treatment of AS may be by reducing the expression of CD36 protein and mRNA in AS rabbit macrophages.[17]

Although there have been randomised controlled trials (RCTs) and controlled clinical trials (CCTs) to study the efficacy of acupuncture intervention on AS, due to sample size and insufficient randomisation methods in clinical trial studies, and the different objectives and outcome evaluation indicators used, the results of the research are not consistent. What type of acupuncture method is better to prevent, stabilise and reverse CAS is still inconclusive. Therefore, this study uses a network meta-analysis (NMA) to summarise the effects of different types of acupuncture treatments on CAS and aims to provide evidence-based medicine for the extensive clinical use of acupuncture as an adjuvant therapy to treat AS.

## OBJECTIVE
The objectives of our study are as follows:
1. To evaluate the efficacy and safety of acupuncture as adjuvant therapy in the treatment of AS.
2. To present the ranking of therapeutic classes.

## METHODS
### Study registration
This NMA will be performed according to the guidelines of the Preferred Reporting Items for Systematic Reviews and Meta-analyses (PRISMA) statement.[18] The protocol has been registered on the PROSPERO website (https://www.crd.york.ac.uk/prospero/) and the PROSPERO registration number is CRD42020207260.

### Patient and public involvement
No patient or member of the public will be involved in the study directly.

### Data sources and search strategy
Seven electronic databases including PubMed, EMBASE, Medline, Cochrane Library, CNKI, Wanfang Database and CQVIP will be searched from inception to December 2020. The search strategy will be adjusted according to the characteristics of the database. The detailed search strategy for PubMed will be shown in table 1, while other databases are shown in online supplemental appendix 1.

### Inclusion and exclusion criteria
#### Types of studies
All RCTs and CCTs will be included if the full text is available; conference proceedings and dissertations might be included if feasible, irrespective of language and

**Table 1** Search strategy for PubMed database

| Number | Search items |
| --- | --- |
| #1 | Atherosclerosis [MeSH] |
| #2 | Atheroscleroses [Title/Abstract] |
| #3 | Atherogenesis [Title/Abstract] |
| #4 | Carotid atherosclerosis [Title/Abstract] |
| #5 | #1 or #2 or #3 or #4 |
| #6 | Acupuncture [MeSH] |
| #7 | Acupuncture Therapy [Title/Abstract] |
| #8 | Acupuncture, Ear [Title/Abstract] |
| #9 | Acupuncture Points [Title/Abstract] |
| #10 | Electroacupuncture [Title/Abstract] |
| #11 | Auricular point [Title/Abstract] |
| #12 | #6 or #7 or #8 or #9 or #10 or #11 |
| #13 | Statins [Title/Abstract] |
| #14 | atorvastatin [Title/Abstract] |
| #15 | aspirin [Title/Abstract] |
| #16 | #13 or #14 or #15 |
| #17 | randomized controlled trial [Publication Type] |
| #18 | randomized [Title/Abstract] |
| #19 | randomly [Title/Abstract] |
| #20 | controlled clinical trial [Title/Abstract] |
| #21 | clinical trial [Title/Abstract] |
| #22 | trial [Title/Abstract] |
| #23 | #17 or #18 or #19 or #20 or #21 or #22 |
| #24 | #5 and #12 and #16 and #23 |

MeSH, Medical Subject Headings.

publication. Retrospective studies, case reports, reviews or studies on the mechanism of action will be excluded.

## Participants

Studies that enrolled patients who were diagnosed with CAS, without serious liver and kidney dysfunction, and patients with tumour; while gender, age, ethnicity and nationality will not be limited. The baseline data of the observation group and the control group are comparable. We will follow the relevant standards of the guidelines for atherosclerotic vascular ultrasound examination.[11]

## Types of interventions

Both the observation group and the control group were treated with conventional western medicine, including aspirin or atorvastatin. The intervention of the observation group was to add acupuncture to the conventional western medicine treatment. Acupuncture therapies included acupuncture, electroacupuncture, auricular acupuncture, warm acupuncture, etc, without restrictions on acupuncture operation methods, acupoint selection, duration of treatment and follow-up period.

## Types of comparison

The control group will include treatments of placebo or the same western medicine treatment in the observation group, such as aspirin or atorvastatin.

## Outcomes

The primary outcome indicator will be the changes of cIMT and VPQ before and after treatment.

The secondary outcomes include the changes of carotid plaque Crouse score, greyscale median and lipid levels. Safety outcomes include the type and frequency of occurrence of adverse reactions, such as subcutaneous haematoma, fatigue, palpitations, etc.

## Selection of studies

Based on the inclusion and exclusion criteria, two independent reviewers (XW and QM) will remove duplicates and unqualified articles after reading the titles and abstract. For articles that meet the inclusion criteria, the full texts will be evaluated for eligibility. If there is any disagreement between the two reviewers, the third reviewer (XY) will make a final decision after discussion. If the data or information is incomplete, the authors will be contacted. The PRISMA flow diagram (figure 1) will be used to show the research screening process.

## Data extraction and management

All reviewers discussed and developed data extraction forms based on the Cochrane Handbook, and two independent reviewers (XW and QM) will extract data from the included studies. The extracted data include general information (author, publication year, journal, etc), participant, sample size, intervention methods, duration, outcomes and follow-up time (table 2). If the data are incomplete, the corresponding author will be contacted.

Multiarm trials will be assigned into two-arm trials to ensure that the results can be combined.[19]

## Risk of bias assessment

Two reviewers (XW and QM) will use the 'risk of bias' tool from the Cochrane Handbook (V.5.1.0) to assess the quality of the included studies.[18 20] The contents of the evaluation include random sequence generation, allocation concealment, and implementation of blinding, incomplete outcome data, selective reports, and other issues. Any inconsistencies will be decided through discussion with the third reviewer (XY). The risk of bias will be classified as 'high risk of bias', 'low risk of bias' or 'unclear risk of bias'.

## STATISTICAL ANALYSIS
### Standard pairwise meta-analysis

Standard pairwise meta-analysis was performed using Stata V.15.0 software. The OR was used as the effect size for count data, the mean difference was used for measurement data and 95% CI was used for interval estimation. According to the Cochrane Handbook, the Mantel-Haenszel $\chi^2$ test and Higgins $I^2$ test will analyse the heterogeneity. A p value of <0.10 or $I^2$ >50% indicates that the heterogeneity is statistically significant, and the source of heterogeneity is analysed. Based on the possible heterogeneity factors, conduct subgroup analysis, and then sensitivity analysis, were carried out to ascertain the reliability of the data, find abnormal studies that lead to significant heterogeneity and analyse the reasons. Heterogeneity also depends on the size of the impact, the direction of the results and the strength of the evidence.

### NMA and network geometry

WinBUGS V.1.4.3 will be used for network statistical analysis,[21] with a random-effects model containing direct comparison and indirect comparison, and the calculation will be performed by the Bayesian Markov chain Monte Carlo methods. The number of iterations is 50 000, the 10 000 is used for annealing to eliminate the influence of the initial value, and sampling starts after 10 001. We calculate the surface under cumulative ranking area to predict the curative effect of each treatment measure and rank it. The result is expressed as a percentage; the larger the value, the better the curative effect of the intervention.[22]

Stata V.15.0 draws the network geometry. The results of different types of acupuncture effects will be demonstrated: the line between the two circles indicates that there is a direct correlation between the two interventions, and the absence of a line means that there is no direct correlation, and only indirect comparisons can be made. The thickness of the line represents the number of direct comparison studies, the area of the circle represents the number of intervention studies evaluated.[23 24] The comparison-adjusted funnel plot

 

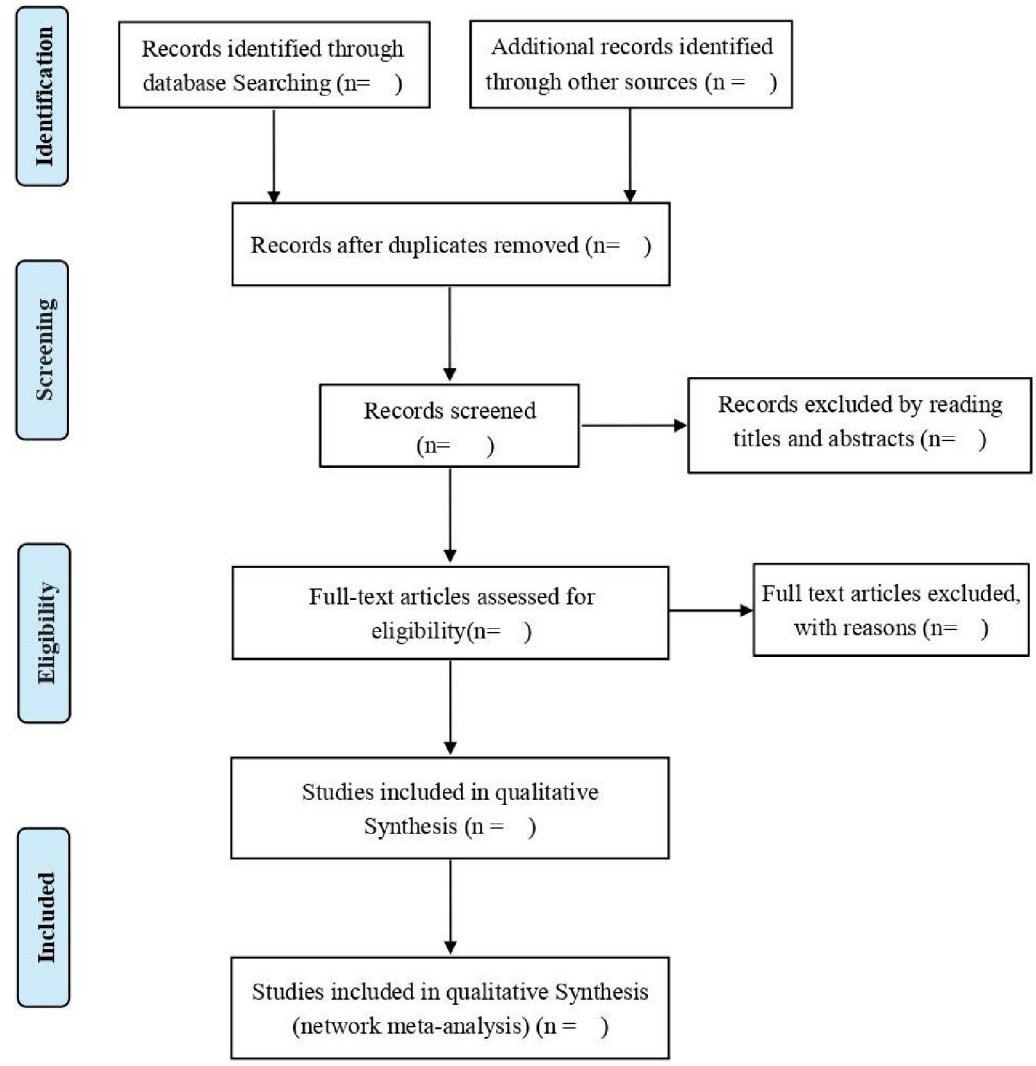

**Figure 1** Preferred Reporting Items for Systematic Reviews and Meta-Analyses flow chart of study selection process.

was used to compare sample effects between studies and evaluate the publication bias of the included studies.

### Assessment of inconsistency

The basic principle of NMA is that the included studies are consistent, and the results of indirect comparison and direct comparison should be similar. We will use loop-specific methods to detect loops of evidence that may have important inconsistencies.[25] A node split model is used to compare the consistency of direct comparison and indirect comparison.[26] P>0.05 means that the consistency between direct and indirect comparisons is better, thus a consistency model is used for analysis. Otherwise,

the inconsistency model is used. For those that have not generated node split, the consistency model is used for analysis.

### Subgroup and sensitivity analyses

To assess the impact of covariates in heterogeneity, such as gender and disease severity, we will explore subgroup analysis.

Subgroup analysis still has obvious heterogeneity, thus sensitivity analysis will be performed. We then eliminate low-quality studies one by one, and then merge the data. The meta-analysis will be repeated and we will compare the results of the two meta-analyses to identify the impact of each study on the overall results.

| | | | Population | | Intervention | | | | Adverse | |
|---|---|---|---|---|---|---|---|---|---|---|
| Reference | Country | Study design | Sample age | Sample size | I | C | Duration | Outcomes | event | Follow-up |

**Table 2** Characteristics of included studies

C, control; I, intervention.

## Assessment of publication biases

If more than 10 trials are included, a Deek's funnel plot will be used to evaluate publication bias. If the angle between the regression line and the x-axis is closer to 90°, it means that there will be less possibility of publication bias.[27]

## Ethics and dissemination

Since the study will be based on de-identified, aggregated published data, ethical approval is not required. The final report of this review will be disseminated through peer-reviewed publications or conference reports.

## DISCUSSION

CAS is a common multiple disease that endangers human health. The common pathological basis of cardiovascular and cerebrovascular diseases has become the main cause of death in the world. Clinically, the treatment of AS is roughly divided into drug and non-drug treatments. Among them, drug treatments include statins, which have antiplatelet, anti-inflammatory and lipid-regulating effects, and inhibit angiogenesis effects in plaques.[28 29] Although great progress has been made in the treatment of AS, the complex and diverse causes and mechanisms of AS (closely related to metabolic, environmental, genetic and other factors) make the existing clinical intervention methods. The treatment of AS is still insufficient, and diseases caused by AS still bring huge health and economic burden to society.[30] In recent years, acupuncture has been widely used as an adjuvant treatment for AS. However, there is no research to rank acupuncture intervention and evaluate the optimal acupuncture method. Therefore, this NMA will conduct a detailed summary and analysis of acupuncture as adjuvant therapy for CAS to determine the optimal acupuncture therapy.

The comparison between therapies in the NMA is based on direct comparisons of interventions within RCTs and indirect comparisons across trials based on a common comparator, such as placebo or some standard treatment. The inter-relationship between multiple interventions can be analysed, and dealt with factors that may have an impact due to the number of included articles, heterogeneity and inconsistency.[31 32]

Nevertheless, this NMA still has limitations. Some low-quality trials may affect the final results of the NMA. In addition, the heterogeneity of different studies may affect the final results of this study. Finally, this study only includes English and Chinese trials, which may lead to the potential risk of ignoring some studies.

Despite the limitations, the latest data should be systematically reviewed. It is concluded that different types of acupuncture can assist patients with CAS to better improve cIMT, and can provide evidence-based medicine for clinical evidence.

**Contributors** XW and QM conceived of the protocol and drafted the manuscript. XW and JL registered the protocol review in the PROSPERO. ZY and XH developed the search strategy. NZ revised the manuscript for methodological and intellectual content. XY contributed to and approved the final manuscript of the protocol review.

**Funding** This study is supported by the National Natural Science Foundation of China (grant number: 82160941, 82160937).

**Competing interests** None declared.

**Patient and public involvement** Patients and/or the public were not involved in the design, or conduct, or reporting, or dissemination plans of this research.

**Patient consent for publication** Not required.

**Provenance and peer review** Not commissioned; externally peer reviewed.

**ORCID iD**
Xianming Wu http://orcid.org/0000-0002-4825-0944

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
