## [Reviewer comments · BMJ Open]

ARTICLE DETAILS

TITLE (PROVISIONAL)	The comparative efficacy of different acupuncture as adjuvant therapy on carotid atherosclerosis: A protocol for systematic review and network meta-analysis
AUTHORS	Wu, xianming; Mo, Qian; Yang, Zhihong; Huang, Xiaolou; Liu, Jiao; Xu, Shuangmei; Zhang, Ning; Yang, Xiaofang

VERSION 1 – REVIEW

REVIEWER	Lu, Liming Guangzhou University of Chinese Medicine
REVIEW RETURNED	02-May-2021

GENERAL COMMENTS	1. In the Types of interventions, is the expression of "moxibustion" and "warm moxibustion" repeated? Or did the author want to express "warm acupuncture"? 2. In the selection of outcome indicators, protocol did not elaborate the consideration of selecting IMT as the primary outcome. Carotid intima-media thickness (cIMT) is associated with the risk of cardiovascular events in the general population, and changes in cIMT are generally considered to be associated with the risk of cardiovascular disease, but has any research reported this correlation — This point needs to be supplemented in the research background. 3. It is suggested that a more detailed explanation should be given on how to evaluate and grade the strength of direct, indirect and network evidence. This network meta-analysis protocol of acupuncture as adjuvant therapy on carotid atherosclerosis is relatively rigorous in structure and logic. The process of writing refers to the requirements of PRISMA, and the items involved are basically complete.
---

REVIEWER	Zhang, Kai Tianjin Gong An Hospital, Acupuncture and Moxibustion
REVIEW RETURNED	28-Jun-2021

GENERAL COMMENTS	As a clinician, I often treat carotid atherosclerosis with acupuncture. The topic is of importance to clinicians and policymaker because the role of unconventional treatments such as acupuncture is still controversial. But there were several points I would like to discuss with the authors. 1. On page 4, lines 46 to 53, The World Health Organization released the National Survey of Non-communicable Diseases in 2018, showing that my country's non-communicable disease....." There is no reference to this sentence. 2. On page 4, lines 54 to 60, "In response to this phenomenon, the United Nations has set a goal and plans to adopt various
---

	measures to make 30 - 70 years old people die prematurely due to cardiovascular disease by 2025. 25% reduction in mortality." This sentence is not very clear. 3. The title of the article is acupuncture. It is mentioned in the literature search that moxibustion is included. Acupuncture and moxibustion are different treatments professionally. 4. The mechanism of carotid atherosclerosis treated by acupuncture should be supplemented. Such as, acupuncture can reduce the expressions of CD36 protein and CD36 mRNA in peritoneal macrophages of atherosclerotic rabbits, which may be one of the mechanisms of EA treatment of atherosclerosis [1]. Study is reported that acupuncture treatment has anti-inflammation, immunity activation and nervous system modulation [2]. [1] Effects of electroacupuncture on expression of CD36 in peritoneal macrophages of rabbits with atherosclerosis. Zhongguo Zhen Jiu. doi: 10.13703/j.0255-2930.2018.02.020. [2] Is acupuncture effective in the treatment of COVID-19 related symptoms? Based on bioinformatics/network topology strategy. Brief Bioinform. doi: 10.1093/bib/bbab110. 5. In the Data sources and search strategy section, It is recommended that authors list search strategies for other databases, as different database search methods are different, and put this section in the supplementary material. 6. The authors searched relevant studies by searching English and Chinese databases. I was wondering if this search strategy might induce publication bias in the overview's conclusion? Did the authors obtain any unpublished meta-analyses (e.g., in conference proceedings, dissertations, etc.). I think such meta-analyses may be still included in this overview to reduce the potential risk of publication bias. 7. There are few studies on acupuncture treatment of carotid atherosclerosis. A related suggestion that the authors focused only on meta-analyses of RCTs, and they excluded meta-analyses that contained studies with designs other than RCTs. Is this exclusion a kind of waste of information? The authors could also present meta-analyses of non-RCTs (with specifying their design types) and downgrade their evidence. 8. Please indicate what are the routine treatments for carotid atherosclerosis. 9. Vessel plaque quantification (VPQ) is preferred for observing plaque morphological characteristics as a non-invasive assessment method. This result is regarded as the primary outcome in some literatures [1]. The authors should use appropriate primary outcome and secondary outcomes on the basis of full search. [1] Acupuncture treatment for carotid atherosclerotic plaques: study protocol for a pilot randomized, single blinded, controlled clinical trial. Trials. doi: 10.1186/s13063-020-04709-0.
--	--

REVIEWER	Chis Ster, Irina St. George's University of London
REVIEW RETURNED	07-Jul-2021

GENERAL COMMENTS	I do not really understand the need or the purpose of the publication of a protocol for a systematic review and/or meta-analysis. This is not a clinical trial so I suggest to carry out the systematic review and the meta-analysis as drafted or planned and following the associated guidelines.
---

VERSION 1 – AUTHOR RESPONSE

Reviewer 1:

Q1: In the Types of interventions, is the expression of "moxibustion" and "warm moxibustion" repeated? Or did the author want to express "warm acupuncture"?

A: As the reviewer said, it is warm acupuncture, which has been corrected.

Q2: In the selection of outcome indicators, protocol did not elaborate the consideration of selecting IMT as the primary outcome. Carotid intima-media thickness (cIMT) is associated with the risk of cardiovascular events in the general population, and changes in cIMT are generally considered to be associated with the risk of cardiovascular disease, but has any research reported this correlation — This point needs to be supplemented in the research background.

A : According to the reviewer's suggestion, we have added the sentence of "Carotid artery intima-media thickness (cIMT) is related to the risk of cardiovascular events in the general population, and cIMT measurement is essential for assessing the risk and incidence of cardiovascular disease" in the introduction.

Q3: It is suggested that a more detailed explanation should be given on how to evaluate and grade the strength of direct, indirect and network evidence.

A: According to the suggestion, the evidence of direct and indirect network analysis has been changed to a more detailed description. The specific details are shown in the MS.

Q4: This network meta-analysis protocol of acupuncture as adjuvant therapy on carotid atherosclerosis is relatively rigorous in structure and logic. The process of writing refers to the requirements of PRISMA, and the items involved are basically complete.

Special thanks to you for your good comments.

Reviewer 2:

Q1: On page 4, lines 46 to 53, The World Health Organization released the National Survey of Non-communicable Diseases in 2018, showing that my country's non-communicable disease....." There is no reference to this sentence.

A: According to the reviewer's suggestion, we have added reference 4. The sentence of "The World Health Organization released the National Survey of Non-communicable Diseases in 2018, showing that my country's non-communicable disease mortality rate is estimated to account for 89% of all deaths, of which cardiovascular disease accounts for 43% of the proportion of non-communicable diseases." were corrected as "The World Health Organization reports that non-communicable diseases cause 41 million deaths each year, equivalent to 71% of the total global deaths. Among them, cardiovascular diseases cause 17.9 million deaths each year."

Q2: On page 4, lines 54 to 60, "In response to this phenomenon, the United Nations has set a goal and plans to adopt various measures to make 30 - 70 years old people die prematurely due to cardiovascular disease by 2025. 25% reduction in mortality." This sentence is not very clear.

A: The sentence of "In response to this phenomenon, the United Nations has set a goal and plans to adopt various measures to make 30 - 70 years old people die prematurely due to cardiovascular

disease by 2025. 25% reduction in mortality.” were corrected as “WHO plans to reduce premature mortality from non-communicable diseases by one-third by 2030 through treatment and prevention.”

Q3: The title of the article is acupuncture. It is mentioned in the literature search that moxibustion is included. Acupuncture and moxibustion are different treatments professionally.

A: We have deleted moxibustion.

Q4: The mechanism of carotid atherosclerosis treated by acupuncture should be supplemented. Such as, acupuncture can reduce the expressions of CD36 protein and CD36 mRNA in peritoneal macrophages of atherosclerotic rabbits, which may be one of the mechanisms of EA treatment of atherosclerosis [1]. Study is reported that acupuncture treatment has anti-inflammation, immunity activation and nervous system modulation [2].

A: According to the reviewer’s suggestion, we have added the sentence of “it has the effects of anti-inflammation, immunity activation and nervous system modulation” and “One of the mechanisms of electroacupuncture treatment of atherosclerosis may be by reducing the expression of CD36 protein and mRNA in AS rabbit macrophages” in the introduction.

Q5: In the Data sources and search strategy section, it is recommended that authors list search strategies for other databases, as different database search methods are different, and put this section in the supplementary material.

A: As reviewer suggested that we have added search strategies for other databases.

Q6: The authors searched relevant studies by searching English and Chinese databases. I was wondering if this search strategy might induce publication bias in the overview’s conclusion? Did the authors obtain any unpublished meta-analyses (e.g., in conference proceedings, dissertations, etc.). I think such meta-analyses may be still included in this overview to reduce the potential risk of publication bias.

A: According to the reviewer’s suggestion, in order to reduce the risk of bias, we revised the types of studies and corrected it to irrespective of language, and conference proceedings, dissertations might be included if feasible.

Q7: There are few studies on acupuncture treatment of carotid atherosclerosis. A related suggestion that the authors focused only on meta-analyses of RCTs, and they excluded meta-analyses that contained studies with designs other than RCTs. Is this exclusion a kind of waste of information? The authors could also present meta-analyses of non-RCTs (with specifying their design types) and downgrade their evidence.

A: According to the reviewer’s suggestion, we have added controlled clinical trials.

Q8: Please indicate what are the routine treatments for carotid atherosclerosis.

A: We have added the sentence of “The current conventional Western medicine treatments for carotid atherosclerosis mainly consist of lifestyle modification, drug therapy, etc. Lifestyle modification includes smoking cessation, control of energy intake, etc.” in the introduction.

Q9: Vessel plaque quantification (VPQ) is preferred for observing plaque morphological characteristics as a non-invasive assessment method. This result is regarded as the primary outcome

in some literatures [1]. The authors should use appropriate primary outcome and secondary outcomes on the basis of full search.

A: As reviewer suggested that we have added vessel plaque quantification as primary outcome, grey-scale median and lipid levels as secondary outcomes in the outcomes section. We have also added the sentence of "Of course, vascular plaque quantification (VPQ) is a non-invasive evaluation method for observing the morphological characteristics of plaques, and it is also regarded as the main observation in clinical practice" in the introduction.

Reviewer 3:

I do not really understand the need or the purpose of the publication of a protocol for a systematic review and/or meta-analysis. This is not a clinical trial so I suggest to carry out the systematic review and the meta-analysis as drafted or planned and following the associated guidelines.

A: Special thanks to you for your good comments. We have referred to the requirements of PRISMA.

VERSION 2 – REVIEW

REVIEWER	Zhang, Kai Tianjin Gong An Hospital, Acupuncture and Moxibustion
REVIEW RETURNED	22-Nov-2021
GENERAL COMMENTS	The authors have fulfilled the given comments and rectified the quality of the manuscript.